# Individual response to antidepressants for depression in adults–a meta-analysis and simulation study

**Klaus Munkholm** [1]\*, **Stephanie Winkelbeiner**[2], **Philipp Homan**[2]

**1** Nordic Cochrane Centre, Rigshospitalet, Copenhagen, Denmark, **2** Psychiatric University Hospital Zurich, University of Zurich, Zurich, Switzerland

\* km@cochrane.dk

## Abstract

### Background

The observation that some patients appear to respond better to antidepressants for depression than others encourages the assumption that the effect of antidepressants differs between individuals and that treatment can be personalized.

### Objective

To compare the outcome variance in patients receiving antidepressants with the outcome variance in patients receiving placebo in randomized controlled trials (RCTs) of adults with major depressive disorder (MDD) and to illustrate, using simulated data, components of variation of RCTs.

### Methods

From a dataset comprising 522 RCTs of antidepressants for adult MDD, we selected the placebo-controlled RCTs reporting outcomes on the 17 or 21 item Hamilton Depression Rating Scale or the Montgomery-Asberg Depression Rating Scale and extracted the means and SDs of raw endpoint scores or baseline to endpoint changes scores on eligible depression symptom rating scales. We conducted inverse variance random-effects meta-analysis with the variability ratio (VR), the ratio between the outcome variance in the group of patients receiving antidepressants and the outcome variance in the group receiving placebo, as the primary outcome. An increased variance in the antidepressant group would indicate individual differences in response to antidepressants.

### Results

We analysed 222 RCTs that investigated 19 different antidepressants compared with placebo in 345 comparisons, comprising a total of 61144 adults with an MDD diagnosis. Across all comparisons, the VR for raw endpoint scores was 0.98 (95% CI 0.96 to 1.00, $I^2 = 0\%$) and 1.00 (95% CI 0.99 to 1.02, $I^2 = 0\%$) for baseline-to-endpoint change scores.

**Data Availability Statement:** All data and code are available online on the Open Science Framework platform (https://osf.io/5gpe4/).

**Funding:** The authors received no specific funding for this work.

**Competing interests:** The authors have declared that no competing interests exist.

## Conclusion

Based on these data, we cannot reject the null hypothesis of equal variances in the antidepressant group and the placebo group. Given that RCTs cannot provide direct evidence for individual treatment effects, it may be most reasonable to assume that the average effect of antidepressants applies also to the individual patient.

## Introduction

Major depressive disorder (MDD) is a common illness [1] and estimated to be the leading cause of disability worldwide by the World Health Organization [2]. Antidepressants, either alone or in combination with psychotherapy, are recommended by guidelines for the treatment of MDD [3–8]. Yet, the effect of antidepressants on depression symptoms is small compared with placebo [9–16], with a difference in depression scores of approximately 2 points [13,17] on the 17-item Hamilton Depression Rating Scale (HAMD) [18] (range 0–52). In addition, considering the methodological limitations of many antidepressant trials [19], the true effect of antidepressants for depression in adults remains uncertain [17].

The small effect of antidepressants over placebo [10] is often believed to result from some patients having substantial benefit, while others have less or no benefit from the treatment [1,20,21]. Thus, a common interpretation of the observed heterogeneity in outcomes among patients treated with antidepressants, whether in clinical practice or in the context of a randomized controlled trial (RCT), is that those with the best outcome, often labelled "responders", differ from those with less favourable outcomes, similarly labelled "non-responders". The implicit assumption is that response is a permanent characteristic of the individual patient, and that the observed variability in outcomes can be ascribed to heterogeneity in the treatment effect of antidepressants.

This perceived heterogeneity has motivated efforts to direct the research and treatment agenda into one of personalized medicine. The aim is to match the individual patient to a therapy that best suits their specific characteristics and condition [22]. Personalized medicine includes the search for potential pharmacogenetic markers [23,24], other biomarkers [25,26], and clinical characteristics [27]. Those are hoped to predict the response to antidepressants and to help identify those patients who are most likely to "respond" to antidepressants. Without such prediction markers, the individual patient may be subject to a trial-and-error process where different antidepressants may be given in succession until the desired outcome is reached [28,29]. Despite decades of research, however, no clinical characteristic and no individual or aggregate biomarker [30] has translated into clinical practice for the guidance of treatment selection.

There may, however, be little reason to assume that a drug that appears to be marginally effective in a larger population, will turn out to be more effective in a subpopulation [31], let alone in certain individuals. The promise of personalized antidepressant treatment may thus rest largely on untested assumptions about individual differences in response to treatment, often in the context of RCTs. Randomized, controlled trials are the gold standard to evaluate the efficacy of a drug compared with placebo. Yet, estimating individual response to treatment, known as the treatment-by-patient interaction, is complex and cannot be inferred from an RCT. The design of an RCT allows to compare the treatment with the control group and with that to estimate an average effect of the treatment. While it is common to uncritically attribute the variation in outcomes observed among patients in RCTs to characteristics of the

individual, the more likely interpretation that these differences in response reflect random or other sources of variation is often not considered [32]. RCTs do not allow to distinguish between individual responses to the treatment and random variability or any scenario in between [32]. To determine on an individual level whether a drug works thus depends on the comparison within patients to phases without the treatment—a counterfactual [31]; therefore, designs such as repeated crossover trials are required [33].

While RCTs cannot be directly used to distinguish individual response to treatment from other components of variation, they may provide indirect evidence about the presence of individual differences in response. This indirect evidence is comprised by the variance of the treatment compared to the control group [33]. An increased variance in the treatment group compared with control could indicate the presence of individual differences in response to antidepressants, known as treatment-by-patient interaction [33–35]. Following this rationale, we recently showed that evidence for a treatment-by-patient interaction in RCTs of antipsychotics compared with placebo in schizophrenia was surprisingly small [36]. Two recent studies applied similar methods and found no evidence of heterogenous treatment effects of antidepressants for depression [37,38]; one of those studies also provided a helpful discussion of the variability ratio as an indicator for treatment effect variability [38].

Thus, we here extended our previous work to antidepressants, investigating whether there is empirical evidence for individual differences in response to antidepressants in RCTs. Additionally, we illustrated the different components of variation in RCTs and crossover trials using simulated data to highlight the component of interest: the treatment-by-patient interaction.

## Materials and methods

### Trial simulation

To illustrate the different components of variation, we simulated an RCT with 30 adults with MDD, informing the parameters by a systematic review and network meta-analysis of antidepressants for depression in adults [10]. Accordingly, patients were randomized to either treatment with the antidepressant sertraline or placebo. Symptom severity was assessed with the 17-item HAMD, with a mean baseline score of 25 points and a mean (standard deviation (SD)) endpoint score of 12.5 (8) points in the treatment group and 14.5 (8) points in the placebo group.

We first simulated the data to demonstrate the variation in effects across both groups and the variation between patients in the treatment group. We illustrated the effect of dichotomization of the treatment group into categories of "responders" and "non-responders", depending on whether the patients' endpoint score decreased by 50% or more compared with baseline [39]. Second, we simulated a crossover trial by adding a placebo condition for the patients in the treatment group to show the consequence of between-patient variation. Third, we simulated the repeated measurement over time to explain the component of within-patient variance that is due to random fluctuation of symptoms. Lastly, we simulated a double crossover trial, to elucidate how such a design allows to separate the treatment-by-patient interaction from other variance components.

### Meta-analysis

**Information sources.** We used the data of a recent meta-analysis of antidepressants for depression in adults, comprising 522 studies and 116 477 patients [10]. The authors of the meta-analysis had searched the Cochrane Central Register of Controlled Trials, CINAHL, Embase, LILACS database, MEDLINE, MEDLINE In-Process, PsycINFO, AMED, the UK

National Research Register, and PSYNDEX from the date of their inception to January 8, 2016. The search was performed with no language restrictions, supplemented with manual searches for published, unpublished, and ongoing RCTs in international trial registers, websites of drug approval agencies, and key scientific journals in the field [10]. Included were double-blinded RCTs comparing antidepressants (provided dosing was within the therapeutic range) with placebo or another antidepressant as oral monotherapy for the acute treatment of adults ($\geq$ 18 years of age, both sexes) with a primary MDD diagnosis [10]. Further, the included antidepressants were second-generation antidepressants approved by the regulatory agencies in the USA, Europe, or Japan, the tricyclics amitriptyline and clomipramine included in the World Health Organization Model List of Essential Medicines, and trazodone and nefazodone, which were included because of their "distinct effect and tolerability profiles" [10]. This data is available online, accompanying the published article [10] (https://data.mendeley.com/datasets/83rthbp8ys/2).

## Eligibility criteria

We applied additional eligibility criteria to this dataset [10] including only placebo-controlled studies that reported the sample size, mean and SD of baseline-to-endpoint change scores or raw endpoint scores on either the HAMD-17, HAMD-21 [18], or the Montgomery-Asberg Depression Rating Scale (MADRS) [40], which were the most frequently used scales among the studies [17].

## Study records, selection process, and data collection

We downloaded the full online dataset (https://data.mendeley.com/datasets/83rthbp8ys/2) and imported it into the statistical software R (version 3.6.0). We selected studies based on the information in the online dataset and refrained from collecting additional data. From the dataset, we extracted information on study identification (e.g. first author, trial registration number), study year, mean (SD) raw endpoint or baseline-to-endpoint change score on the HAMD-17, HAMD-21, and MADRS, and sample size.

## Risk of bias in individual studies and across studies

We did not perform assessment of the risk of bias in the included studies but instead referred to the assessment accompanying the dataset [10] which stated using the Cochrane risk of bias tool.

## Statistical analysis

The SDs of the baseline-to-endpoint change scores or raw endpoint scores of the antidepressant and placebo group include the same variance components. The antidepressant group may, however, in addition contain a possible treatment-by-patient interaction, indicating individual response differences. A different variance in the antidepressant group compared with placebo thus would indicate the presence of a treatment-by-patient interaction. To test this hypothesis, we calculated the relative variability of the antidepressant and placebo group for each study as the log variability ratio (log VR) [41] with

$$logVR = log\left(\frac{SD_{tx}}{SD_{ct}}\right) + \frac{1}{2(N_{tx} - 1)} - \frac{1}{2(N_{ct} - 1)}$$

where SD is the reported sample SD for the treatment (tx) and the control (ct) group, and N the respective sample size [35]. We further calculated the corresponding sampling variance,

$SD^2_{logVR}$, for each comparison of an antidepressant with placebo with

$$SD^2_{logVR} = \frac{1}{2(N_{tx} - 1)} + \frac{1}{2(N_{ct} - 1)}$$

As the log VR assumes normality of the underlying data, we checked for indications of skew of endpoint scores by calculating the observed mean minus the lowest possible value and dividing by the SD [42]. A ratio less than 2 is suggestive of skew, whereas a ratio less than 1 is strong evidence of skew [43].

We weighted the log VR with the inverse of its corresponding sampling variance [44] and entered it into a random-effects model using restricted maximum-likelihood estimation. We back-transformed the results to obtain a VR that indicates greater variability in the antidepressant group compared with placebo for values greater than 1 and less variability in the antidepressant group compared with placebo for values smaller than 1.

For studies that investigated multiple antidepressants or doses compared to placebo, we divided the sample size of the placebo group by the number of treatment arms while retaining the mean and SD and creating multiple pair-wise comparisons for those studies.

Our primary outcome was the overall summary estimate for the VR across all included comparisons for raw endpoint scores and baseline to endpoint change scores, respectively. We conducted subgroup analyses by (1) type of antidepressant and (2) symptom severity scale. We checked the robustness of our meta-analyses to indications of skew in the included studies in sensitivity analyses. Between-study heterogeneity was assessed using the $I^2$ statistic.

## Data and code availability

All analyses were performed with the R packages metafor, version 2.1.0 [44], and meta, version 4.9.6 [45]. All data and code are available online on the Open Science Framework platform (https://osf.io/5gpe4/). This study and protocol were pre-registered on the Open Science Framework platform (https://osf.io/u4c6p). The protocol is also made available as supporting information.

## Results

### Simulation study

Plotting the baseline-to-endpoint change scores for both the treatment and control group of an RCT, we may observe differences in response to the antidepressant (Fig 1A). Ignoring the variation in outcomes in the control condition and focusing on the variation between individuals in the treatment condition only, we may rank patients according to their outcome and infer that some patients responded better to the treatment than others (Fig 1B). This perception may be strengthened when further dichotomizing the patients in the treatment group as either "responders" or "non-responders" (Fig 1C).

Adding a simulated crossover condition to the initial RCT (Fig 1A) allows the within-patient comparison between the antidepressant and the placebo condition. Only by comparing the response to the antidepressant to the response to placebo within a patient is a comparison of the effect of the treatment across patients, rather than just the observed outcome possible. Inferences about whether the antidepressant effect is constant across patients (Fig 2A and 2B) or whether the effect of antidepressants is different in each patient (Fig 2C and 2D) are now possible.

In addition to the between-patient variation, RCTs include yet another component of variation: the within-patient variation. Symptoms may fluctuate randomly over time within a

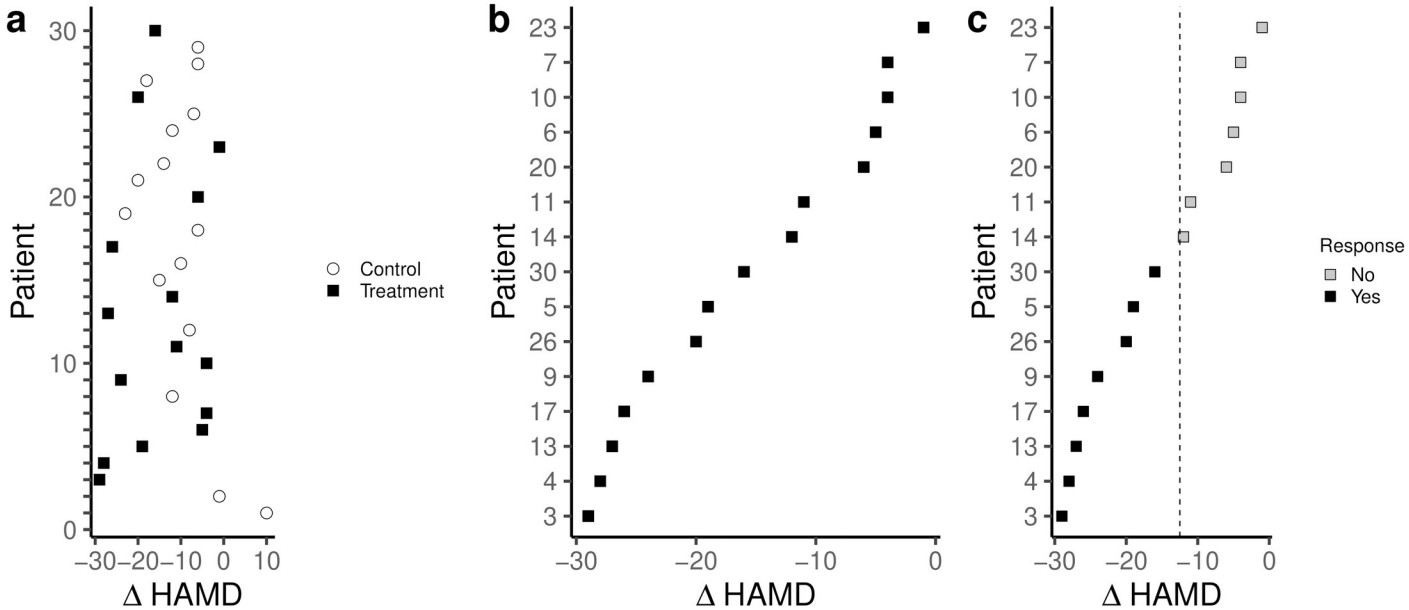

**Fig 1. Simulation of an RCT with 30 patients with MDD randomized to antidepressants or placebo with a mean difference between groups of 2 points on the HAMD-17.** Plotting the observed outcomes of both the antidepressant and placebo group shows variation and overlap in the outcomes in the two groups (a). Observing only the variation in the outcomes in the antidepressant group, while ignoring the variation in the placebo group, and ranking patients according to their outcome can encourage the assumption that the effect differs between individuals (b). Dichotomizing the outcomes based on arbitrary thresholds and the subsequent categorizing of patients as "responders" and "non-responders" can accentuate the perception of individual differences in treatment response and tempt into interpreting those as stable characteristics of the individual patient (c).

patient (Fig 3A). While a larger variation may be observed in some patients compared with others, all might have had the same mean symptom severity when averaging observations over time (Fig 3B). It is thus possible that the within-patient variation alone can explain the variation in the observed outcome in an RCT as well as the variation in the net-benefit observed in a single crossover trial.

While a simple crossover trial accounts for between-patient variation in effects, it does not provide information on whether the observed effect in a given patient is a constant feature of that individual. This is only possible by repeating the crossover trial (Fig 1 in S1 File) [33,36].

In a repeated crossover study, equal variances would be observed in case of a constant treatment effect across all patients; a consequence of a constant effect is that the treatment does not affect variability (Fig 4A). An increased variance in the treatment condition relative to the control condition may arise in one of two different scenarios. One scenario is the existence of two subpopulations with different responses to the antidepressant, a treatment-by-subgroup interaction (Fig 4B). Such a scenario would call for stratified medicine, in which treatment with an antidepressant is conditional on the patient belonging to a subgroup of individuals that share certain characteristics. An alternative scenario resulting in increased variance in the observed effect in the treatment condition compared with the control condition is the existence of a variable effect in each individual, a treatment-by-patient interaction, without any subgroup sharing a common effect (Fig 4C). Such a scenario would call for personalized medicine in which the antidepressant would be conditional on features unique to the individual.

How differences in the variances of the antidepressant and placebo group will be reflected by the VR depends on the distribution of the individual response to antidepressant and the magnitude of that response. The VR will be higher in situations where more individuals show

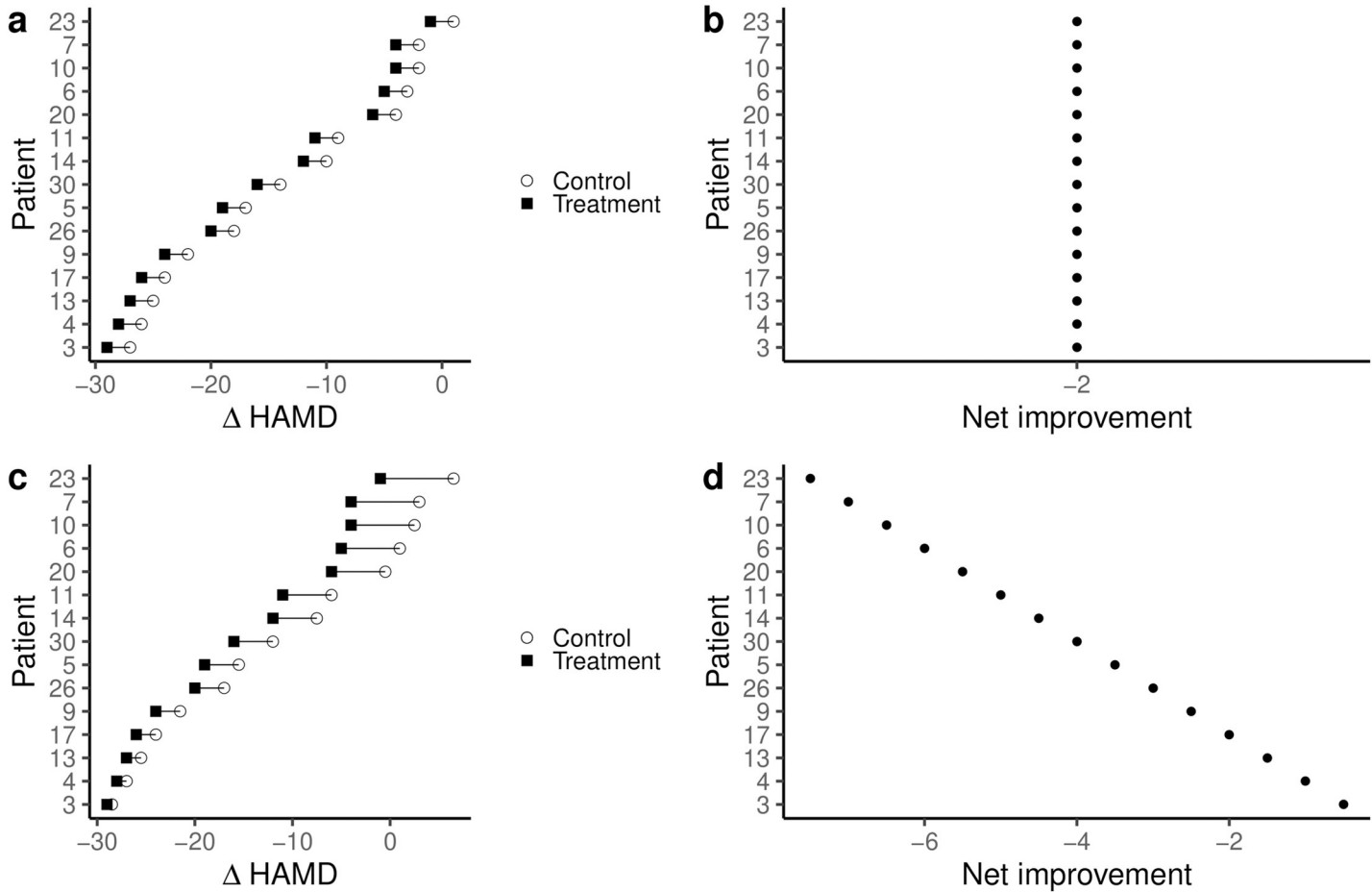

**Fig 2. Simulation of adding a single crossover condition to the initial RCT shows that inferences about individual treatment effects can be misleading when only considering the observed outcomes.** When adding a crossover condition, it is possible that the outcomes observed in the patients under the antidepressant condition would be paralleled by the outcomes under a crossover placebo condition (a). In such a scenario, the net benefit, the actual effect of the treatment, is constant between individuals (b). Patients classified as "responders" when just observing their outcome under the treatment condition (Fig 1C) would thus not differ from those classified as "non-responders". It is also possible, that the patients with the best outcome under the initial treatment condition would also have the best outcomes in the crossover control condition (c). In such a scenario, the patients with the best outcome would, in fact, experience the smallest net benefit of the treatment (d). Those patients that would otherwise be classified as "non-responders" based on their observed outcomes (Fig 1C) would experience the largest net benefit.

a stronger than average response to antidepressants, and smaller in situations where fewer individuals show a less strong response to antidepressants (Fig 2 in S1 File).

## Meta-analysis of empirical data

The initial dataset comprised 522 studies, of which we included only the 304 that were placebo-controlled. Of these, we excluded 46 studies as they did not apply any of the HAMD-17, HAMD-21 or MADRS scales. We excluded a further 36 studies as they did not have complete outcome data. We thus included 345 comparisons from 222 RCTs that investigated a total of 19 antidepressants (see S2 File for list of included studies). The studies included a total of 61144 patients of which 38254 patients received an antidepressant and 22890 received placebo. Details of the included studies are available in the Tables 1 and 2 in S1 File. Of the total dataset of 522 studies, 9% were rated as high risk of bias, 73% as "moderate" and 18% as low risk of bias [10]. Due to limitations in their risk of bias assessment [17] it was not possible to extract the assessment for only the studies included in our analysis. There was some indication of

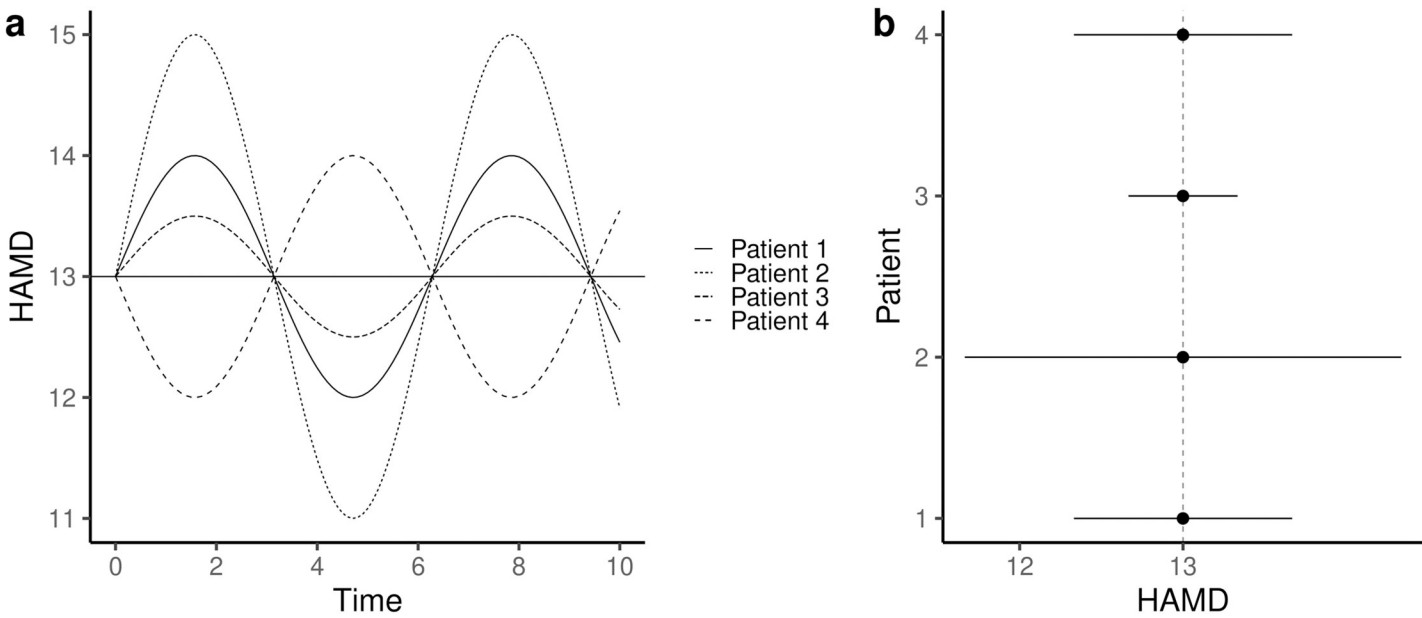

**Fig 3. Simulation of repeated measurements of depression symptom severity on the HAMD-17 among 4 patients of the initial RCT simulation (Fig 1A).** The HAMD-17 score may fluctuate over time independently of the intervention (a). While the mean HAMD-17 score may be the same across the patients, the amount of variation may differ (b). In such a scenario, within-patient variation may account for the entire variation in the observed outcome in an RCT.

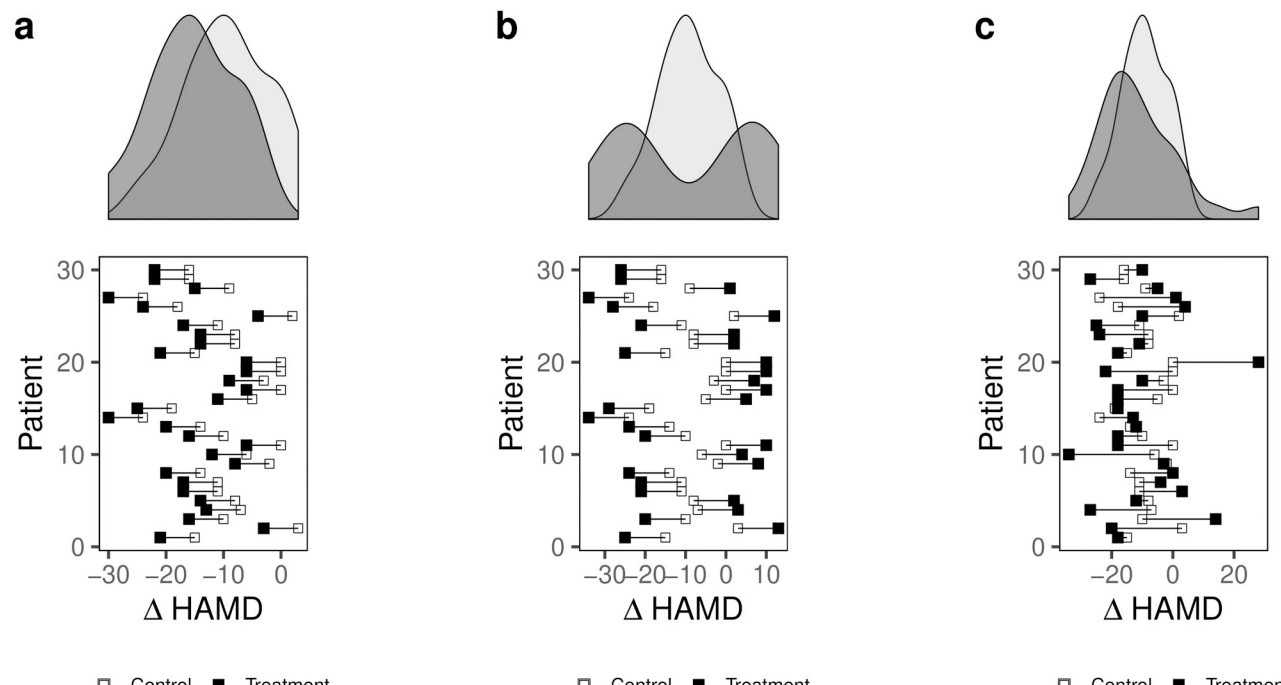

**Fig 4. Simulation of a crossover trial of 30 patients with MDD randomized to subsequently receive an antidepressant and placebo with a mean improvement of 2 points on the HAMD-17.** The marginal density plots illustrate the distribution of outcomes in the two conditions in three different scenarios: (a) illustrates a scenario with a constant treatment effect, in which the variances in the two groups would be equal; (b) shows a scenario with two subpopulations with different effects. Although the effect is the same for all patients in each subgroup, the distribution in the treatment condition has higher variability than the placebo condition; (c) shows a scenario with a variable effect in each patient in which greater variability is observed in the treatment condition.

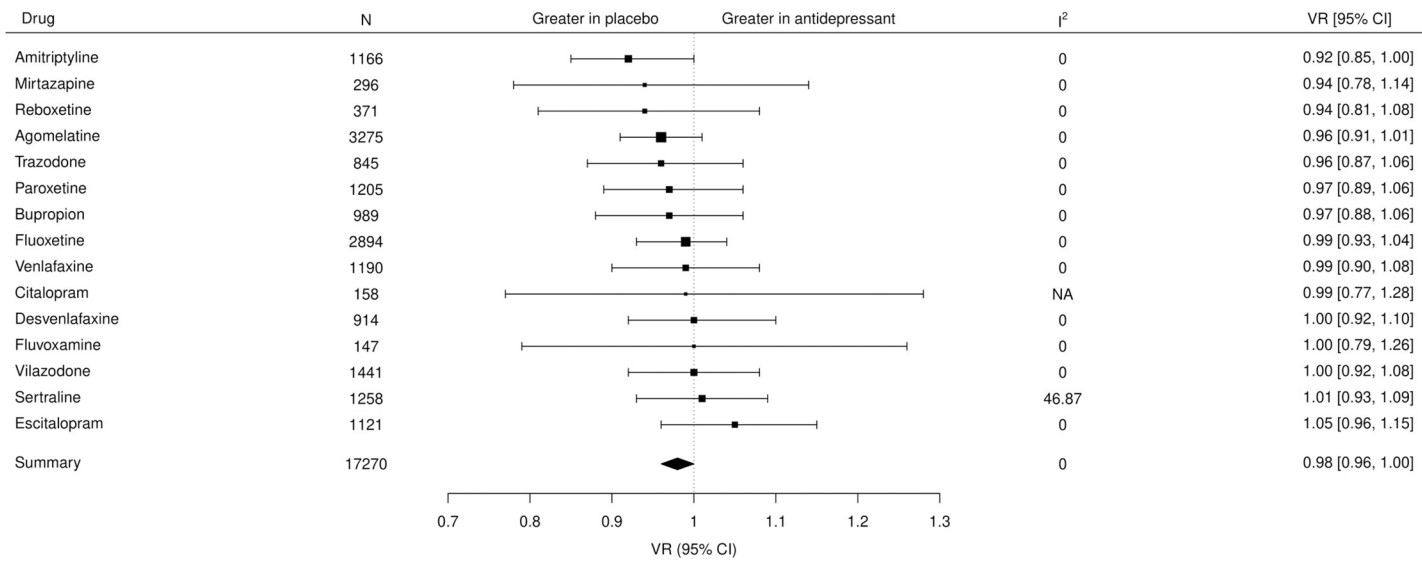

**Fig 5.** (A) Forest plot of the VR for each antidepressant versus placebo for endpoint scores and for (B) baseline-to-endpoint change scores. VR: Variability ratio; I²: inconsistency (%); CI: confidence interval.

skew in most study arms (Fig 3 in S1 File). Strong evidence of skew was observed in five of the 179 arms with endpoint scores (Table 3 in S1 File).

Across all comparisons, we found no difference in variance of the antidepressant and placebo group for either raw endpoint scores ($VR$ = 0.98, 95% confidence interval (CI) 0.96 to 1.00, $I^2$ = 0%) or baseline to endpoint change scores ($VR$ = 1.00, 95% CI 0.99 to 1.02, $I^2$ = 0%).

In the subgroup analysis by antidepressant, there was no difference in variance between any antidepressant and placebo (Fig 5A) and no difference in VR between drugs (df = 14, $P$ = 0.91) for endpoint scores and similarly no difference in VR between drugs for change scores

(df = 18, $P$ = 0.99) (Fig 5B). Moderate inconsistency ($I^2$ = 46.9%) was observed for studies of sertraline in the analysis of endpoint scores, which appeared to be due to a single study [46]. After excluding the study, no inconsistency was observed in the meta-analysis of sertraline compared with placebo (Fig 4 in S1 File). The inconsistency observed for studies of escitalopram in the analysis of change scores ($I^2$ = 87.9%), appeared to be due to a single study [47] with a very large effect. After excluding this study [47] no inconsistency was observed in the meta-analysis of escitalopram compared with placebo (Fig 5 in S1 File).

In the subgroup analysis by rating scale, we found no difference in variance of the antidepressant and placebo group for any of the studies reporting on the HAMD-17, HAMD-21 or MADRS, respectively, with no difference in VR between the groups for either endpoint scores (df = 2, $P$ = 0.38; Table 4 in S1 File) or change scores (df = 2, $P$ = 0.90; Table 5 in S1 File).

In the subgroup analysis according to indicators of skew in endpoint score data, there was no difference in VR between categories indicating evidence of skew, suggestion of skew and no skew, respectively (df = 2, $P$ = 0.68). In a sensitivity analysis excluding studies with strong evidence of skewed data, there was no difference in variance between antidepressants and placebo ($VR$ = 0.98, 95% CI 0.96 to 1, $I^2$ = 0%). When analysing only studies with no indication of skew, we observed no difference in variance between antidepressants and placebo ($VR$ = 0.98, 95% CI 0.96 to 1, $I^2$ = 0).

## Discussion

This study investigated the empirical evidence from RCTs for the presence of a personal element of response in adults with MDD by comparing the outcome variance in the antidepressant group with the outcome variance in the placebo group in 222 placebo-controlled RCTs from the last 40 years. Based on these data we cannot reject the null hypotheses of equal variances in the antidepressant group and the placebo group. Further, we illustrated that inferences made based on the observed outcomes in RCTs can be misleading and can encourage the assumption of individual differences in the response to antidepressants.

Our finding of a lack of empirical support for the presence of individual differences in treatment response is in accordance with two recent meta-analyses of the VR in placebo-controlled studies of antidepressants for depression in adults utilising the same dataset of studies, although of different subsets, as we used for our analysis [37,38]. Plöderl et al. [37] used a frequentist approach like ours while Volkmann et al. [38] employed a Bayesian approach. In contrast to our study, which included a total of 222 trials that reported either baseline to endpoint change scores or raw endpoint scores on the three most used depression symptoms severity rating scales in the dataset, Plöderl et al. [37] and Volkmann et al. [38] both included 169 trials using any rating scale in the subset of trials reporting baseline to endpoint change scores. Plöderl et al. [37], although only made available in their supplement, additionally analysed endpoint scores based on 84 studies. As opposed to our approach, both studies [37,38] analysed the coefficient of variation (CVR), in addition to the VR, as a variability effect size and found a lower summary CVR in the antidepressant group compared with the placebo group. Different requirements and assumptions apply to the VR and the CVR. Both assume normality of the data, although for the CVR, adjustments to the calculation can make the method suitable for non-normally distributed data [48]. The CVR, additionally, requires the data to be measured on a ratio scale [48]. However, given the psychometric properties of the HAMD and the MADRS, and likely other depression severity rating scales, they should not be considered ratio scales. Thus, while some items on the HAMD measure single symptoms along a meaningful continuum of severity, many do not; for some items there is no clear ordering of variables, leading to ordinal and nominal scaling being combined in single items [49]. Further, as the

scoring of individual items is not necessarily related to the severity of depression and as items do not contribute equally to the total score [49], the distance between two points on the scale does not reflect a constant difference in depression severity. As both the HAMD [49,50] and the MADRS [50] are multidimensional the interpretation of a single summed score is unclear [49] and changes over time because of a lack of longitudinal measurement invariance of the scales [51]. Therefore, investigating variance using the CVR as effect size in studies using the HAMD and the MADRS, as done in previous studies [37,38], does not comply with the requirements for such analysis, regardless of whether endpoint- or change scores are analysed [48]. Another issue related to the CVR is the necessity of addressing the relationship between the mean and the standard deviation in the analysis, as spurious findings may otherwise be observed [38]. In addition to these issues, none of the studies [37,38] pre-registered a study protocol. During peer-review, a third study involving a meta-analysis of the VR based on a cohort of 28 industry-sponsored trials of selective serotonin reuptake inhibitors (SSRIs) for depression was published online [52]. The study, which was not based on a pre-registered protocol, in accordance with our findings, found no difference in the variance of the HAMD-17 endpoint scores between patients receiving SSRIs and patients receiving placebo. Taken together, the similar findings in three other studies using overlapping but different study populations and various methodologies, may be taken as one indicator of the robustness of our findings.

## Strengths and limitations of the study

Strengths of our study include robust design and transparency. We prespecified the research question and prospectively registered a protocol for the study and. Deviations from the protocol are described in the S1 File. We used the largest available dataset of placebo-controlled randomised studies of antidepressants and have shared all our data and analysis code. Our simulations provide graphical illustration of important limitations in in the inferences that can be made from RCTs.

There are limitations to our meta-analysis. First, we relied on previously extracted data from individual studies [10] and did not extract the data ourselves. It may be considered a limitation that we did not perform or perform risk of bias assessment on the included studies, as the risk of bias may be higher than initially reported [17]; it is unclear, however, how risk of bias in the included studies would affect the variability ratio. Second, the trials were overall of short duration and antidepressant trials generally apply strict inclusion criteria; it is possible that individual treatment effects could be observed with longer trial duration and in populations that differ from those usually included in clinical trials. Third, the HAMD has psychometrical limitations [49] and the variability ratio could potentially differ when analysing other outcomes. Fourth, a large proportion of the data showed indication of asymmetrical distributions. While our subgroup- and sensitivity analyses did not indicate that the VR differed according to the indication of skew in the data, it cannot be excluded that the distribution of data influenced our results. Fifth, the search for studies included in the dataset was conducted in 2016. However, given our findings and the large dataset we used, it could be considered unlikely that adding expectedly relatively few additional studies through an updated search would change our results. Finally, although an increase in the variance in the antidepressant group compared with placebo would indicate a treatment-by-patient interaction, the reverse is not true—equal variance in the antidepressant and the placebo group does not eliminate the possibility of a treatment-by-patient interaction. However, such a scenario can only be observed in a situation where the treatment effect variance and the variance of the placebo condition are correlated, and their covariance equals exactly half the negative value of the

treatment effect variance. While theoretically possible, it could be argued that it is not very likely. Thus, the most parsimonious explanation for equal variances in the treatment and placebo group is that of a constant effect [34].

## Implications of the study

Translating the population average effect of an intervention into an effect for the individual patient requires additional assumptions when based on an RCT; one is that of a constant effect, that would make the results relevant to every patient [53]. One consequence of a constant effect is that the treatment does not affect the outcome variance, which would therefore be expected to be equal in the treatment and placebo groups [34]. Studying the variance in the treatment and placebo groups of an RCT thus provides the means for quantifying differences in variability that may arise due to the investigated treatment [35,36]. This approach allows for inferences about a potential treatment-by-patient interaction. Thus, increased variability in the antidepressant group compared with the placebo group could arise due to individual differences in response to treatment [34]. An alternative explanation would be the presence of two subgroups [34]: one subgroup of patients having a small or even negative effect of treatment with antidepressants, another subgroup having a large effect of the treatment. Such a scenario would call for a stratified research design to identify the subgroup most likely to benefit from the treatment with antidepressants and the subgroup in which such a treatment would have no or even a negative effect, and for the subsequent practice of stratified medicine rather than personalized medicine. An analysis of the variability ratio does not allow for drawing conclusions on the individual level, and thus cannot distinguish between a potential treatment-by-patient interaction and that of a treatment-by-subgroup interaction. Yet, our results showing that the variance in the antidepressant group did not differ from the placebo group, may indicate that neither of those scenarios are likely.

Careful appreciation of the variance components of trials that are discernable by analysing RCTs is necessary to recognize which conclusions can be derived from them. While an RCT can ideally provide an unbiased estimate of the effect of treatment by comparing the outcomes of the patients in the treatment group to the outcomes of the patients in the placebo group, it cannot inform on variation in effects between patients [54]. Without knowledge about how the same patient would have fared under placebo, assumptions about individual differences in response to antidepressants are premature. Therefore, observed outcomes of the treatment group in an RCT alone cannot provide any evidence for a potential treatment-by-patient interaction; rather, focusing on differences in observed outcomes of those receiving treatment can lead to the potentially misleading interpretation that some patients benefit more from the treatment than others, by ascribing all or much of the variation in their observed outcomes to characteristics of the individual. However, the labelling of patients in RCTs as "responders" or "non-responders", apart from the problems with dichotomizing continuous outcomes [55], is problematic. It alludes to the existence of individual response to treatment that cannot be inferred from RCTs to begin with [31]. The often-held viewpoint that the average treatment effect constitutes a simplified summary estimate of a range of responses to the treatment [21,22,56] neglects the fact that RCTs cannot inform on the variation within and between patients [33]. Contrary to the assumption that individual treatment response is a permanent feature of a patient, treatment response may rather, and perhaps most likely, vary from occasion to occasion [31].

Direct evidence of a treatment-by-patient interaction can be provided by repeated crossover trials, including N-of-1 trials. Yet, such trial designs are not without challenges and are often impossible to conduct in conditions such as MDD, where depressive symptoms often

fluctuate, treatment effects may appear only after weeks of treatment, carry-over effects may arise if long-term changes to brain neurochemistry persist [57], and withdrawal effects may be prolonged [58]. However difficult, without the efforts to undertake these designs and analyses, no clear understanding about the presence of a treatment-by-patient interaction is possible. Given that we did not find empirical support for such treatment-by-patient interaction in response to antidepressants, there may be little reason to assume that some patients treated with antidepressants for depression will have a larger effect than the average effect demonstrated in RCTs.

## Conclusions

In our analysis of the variability ratio in RCTs we did not find evidence for individual differences in treatment effects of antidepressants for depression in adults. Given that RCTs cannot provide direct evidence for individual treatment effects, our findings suggest that it may be most reasonable to assume that the average effect of antidepressants applies also to the individual patient.

## Supporting information

**S1 Checklist. PRISMA 2009 checklist.**
(DOC)

**S2 Checklist. PRISMA 2009 flow diagram.**
(PDF)

**S1 Protocol. Study protocol.**
(PDF)

**S1 File.**
(PDF)

**S2 File. List of included studies.**
(PDF)

## Author Contributions

**Conceptualization:** Klaus Munkholm, Stephanie Winkelbeiner, Philipp Homan.

**Data curation:** Klaus Munkholm.

**Formal analysis:** Klaus Munkholm.

**Methodology:** Klaus Munkholm, Stephanie Winkelbeiner, Philipp Homan.

**Project administration:** Klaus Munkholm.

**Writing – original draft:** Klaus Munkholm.

**Writing – review & editing:** Klaus Munkholm, Stephanie Winkelbeiner, Philipp Homan.

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
