## [Decision Letter · Decision Letter 0]

14 Jul 2020

PONE-D-20-17299

Individual response to antidepressants for depression in adults - a meta-analysis and simulation study

PLOS ONE

Dear Dr. Munkholm,

Thank you for submitting your manuscript to PLOS ONE. After careful consideration, we feel that it has merit but does not fully meet PLOS ONE’s publication criteria as it currently stands. Therefore, we invite you to submit a revised version of the manuscript that addresses the points raised during the review process.

**First I would like to thank the 2 reviewers for their important insights. Thank you also for being so fast. **

Then I would like to congratulate your for this nice piece of work. I really appreciate that it was registered a priori and that all the material needed to reproduce the results is transparently being shared. 

Please respond to all the reviewer's comments, and add a few words in your discussion about the overlap between you study and the two others, including some work about the overlap in terms of included studies and its possible implications.

We look forward to receiving your revised manuscript.

Kind regards,

Florian Naudet, M.D., M.P.H., Ph.D.

Academic Editor

PLOS ONE

Journal Requirements:

2. We noted in your submission details that a portion of your manuscript may have been presented or published elsewhere.

"An earlier version of the manuscript has been previously published as a preprint in Psyarxiv (doi: 10.31234/osf.io/m4aqc).

The present manuscript is not currently under for publication by any other journal and has not been published in any other form elsewhere."

Please clarify whether this publication was peer-reviewed and formally published. If this work was previously peer-reviewed and published, in the cover letter please provide the reason that this work does not constitute dual publication and should be included in the current manuscript.

Reviewers' comments:

Reviewer's Responses to Questions

**Comments to the Author**

1. Is the manuscript technically sound, and do the data support the conclusions?

Reviewer #1: Yes

Reviewer #2: Yes

2. Has the statistical analysis been performed appropriately and rigorously? 

Reviewer #1: Yes

Reviewer #2: Yes

3. Have the authors made all data underlying the findings in their manuscript fully available?

Reviewer #1: Yes

Reviewer #2: Yes

4. Is the manuscript presented in an intelligible fashion and written in standard English?

Reviewer #1: Yes

Reviewer #2: Yes

5. Review Comments to the Author

Reviewer #1: This is a well written and timely study. Although the research question was previously addressed by two other groups (Volkmann et al; Plöderl and Hengartner) I think this study is important as, first, it adds an insightful simulation analysis, and second, consistently replicates the findings of the previous studies. Having that said, I want to declare that I am one of the authors of these studies (M. Hengartner).

I have a few suggestions:

The authors may want to add on page 5 that the search for biomarkers and clinical predictors of differential treatment response has consistently failed thus far, which is clearly at odds with the prevailing belief that patients respond differently to antidepressants. Given that equal variance ratios do not exclude the possibility that treatment effect heterogeneity exists, as the authors also rightly state in the discussion, the lack of robust treatment effect modifiers, despite decades of research and millions spend on the search for it, speaks volumes in my opinion. The persistent claim that differential treatment response exists should be supported by evidence. Given that there is no consistent evidence for this claim, it would be more sensible to assume that there is no treatment effect heterogeneity, unless proven otherwise. That so many people firmly stick to this view probably indicates that most researchers and physicians conflate observed treatment outcomes with treatment effects.

The study by Maslej et al (ref 36) was retracted after we pointed out in a letter to the editor that their analysis was flawed. See the retraction note here (https://jamanetwork.com/journals/jamapsychiatry/fullarticle/2767242) and our letter to the editor here (https://www.researchgate.net/publication/342083346_Commentary_on_Maslej_et_al_No_evidence_of_individual_differences_in_response_to_antidepressants). This reference thus needs to be removed.

The authors claim that Volkmann et al as well as Plöderl and Hengartner did not address important methodological issues. I assume that they refer to the simulation study included in their manuscript. If so, they should specify. Otherwise, I don’t really see which methodological issues Munkholm et al addressed that were not addressed in the other two studies. In fact, in my view Volkmann et al. addressed various methodological issues that Munkholm et al. did not consider. For instance, Volkmann et al. empirically addressed the important question, why the VR is the more appropriate method than the CVR, which was erroneously applied in the now retracted paper by Maslej et al. Volkmann et al also empirically tested, how likely treatment-by-patient interaction would be if VR=1.

Shortly after publication of our analysis (Plöderl and Hengartner), we were attacked by Fredrik Hieronymus on Twitter that our study, and by consequence also the present study by Munkholm et al., was severely flawed as it was based on trial-level data instead of individual-patient data. A few days ago, Hieronymus et al now published their own analysis based in IPD data (“Individual variability in treatment response to antidepressants in major depression: comparing trial-level and patient-level analyses”; Acta Psychiatrica Scandinavica, doi: 10.1111/ACPS.13205). To be honest, I don’t really know what to make of the Hieronymus study, as their methodological approach seems rather arbitrary and was not prespecified in a protocol. They corrected the placebo endpoint scores to match the mean scores in the active group and then compared the two distributions with a Kolmogorov-Smirnov test, which, as is well established, will almost always yield a statistically significant result when sample size is large. I also wonder how valid it is to artificially change the placebo endpoint scores, as there is no clear rational, what endpoint score an individual placebo recipient would have achieved had he/she received the antidepressant. The approach chosen by Hieronymus et al is just one possible option among many, and a different approach of course would have produced a different distribution. In any case, I suggest Munkholm et al. have a critical look at this paper and comment on it in the manuscript.

On page 17 the authors discuss in detail the CVR. They make legitimate points, but in my view the main issue why the CVR is inappropriate is the assumption of linear association between the natural logarithm of the mean and the natural logarithm of the standard deviation with a slope coefficient of 1. As comprehensively shown by Volkmann et al., this assumption is severely violated, since the slope coefficient is only about 0.1. The VR assumes a slope coefficient of 0, thus has a much better fit to the data than the CVR. This should be added to the text.

Reviewer #2: In the manuscript “Individual response to antidepressants for depression in adults - a meta-analysis and simulation study”, Munkholm et al. conducted a simulation study and inverse variance random-effects meta-analysis in order to compare the outcome variance in patients receiving antidepressants with the outcome variance in patients receiving placebo in randomized controlled trials of adults with major depressive disorder, as an indicator of individual differences in response to antidepressants. The authors found that there were no differences in variability ratios between across antidepressant vs. placebo comparisons, a result that indicates that the null hypothesis of equal variances cannot be rejected. In the opinion of this reviewer, the authors’ description of methods and results are comprehensive and the paper as a whole a strong contribution to the literature. The research has many strong points, such as: a) a pre-registered protocol, b) the use of a comprehensive dataset from which 222 studies comparing antidepressants vs placebo were extracted, c) all data and code are openly shared on the Open Science Framework, d) the use of relative variability as an indicator for treatment-by-patient interaction, e) conducting a simulation study to investigate possible components of variation in randomized controlled trials and, many more. The results of the simulation study are especially informative.

On a final note, although the database that was used both for informing the simulation study and to conduct the meta-analysis is comprehensive and encompasses many trials, the search for RCTs ended in 2016. As a personal curiosity, although it is clearly stated in the protocol that no supplementary data would be used, it would be very interesting how the results would change if the authors actualized the search for the period 2016-2020 and integrated those results also.

6. PLOS authors have the option to publish the peer review history of their article (what does this mean?). If published, this will include your full peer review and any attached files.

Reviewer #1: **Yes: **Michael P. Hengartner

Reviewer #2: **Yes: **Liviu-Andrei Fodor

---

## [Author Response · Author response to Decision Letter 0]

21 Jul 2020

Response to reviewers

We would like to thank the reviewers and the Editor for their work with reviewing our manuscript and for their very constructive and valuable comments and suggestions. 

We have addressed all the editorial comments and the reviewers’ comments in a point-by-point manner below and, based on the Editor’s suggestion, elaborated on our discussion of the overlap between our study and comparable studies and the potential implications of this overlap. 

Editorial comments 

Then I would like to congratulate you for this nice piece of work. I really appreciate that it was registered a priori and that all the material needed to reproduce the results is transparently being shared. 

Please respond to all the reviewer's comments and add a few words in your discussion about the overlap between you study and the two others, including some work about the overlap in terms of included studies and its possible implications.

Response: We thank the editor for the positive feedback. We have now revised the discussion section and added some text regarding the overlap between our study and similar studies, including overlap in terms of included data. First on p. 16-17: “Our finding of a lack of empirical support for the presence of individual differences in treatment response is in accordance with two recent meta-analyses of the VR in placebo-controlled studies of antidepressants for depression in adults utilising the same dataset of studies, although of different subsets, as we used for our analysis [37,38]. Plöderl et al. [37] used a frequentist approach like ours while Volkmann et al. [38] employed a Bayesian approach. In contrast to our study, which included a total of 222 trials that reported either baseline to endpoint change scores or raw endpoint scores on the three most used depression symptoms severity rating scales in the dataset, Plöderl et al. [37] and Volkmann et al. [38] both included 169 trials using any rating scale in the subset of trials reporting baseline to endpoint change scores. Plöderl et al. [37], although only made available in their supplement, additionally analysed endpoint scores based on 84 studies. As opposed to our approach, both studies [37,38] analysed the coefficient of variation (CVR), in addition to the VR, as a variability effect size and found a lower summary CVR in the antidepressant group compared with the placebo group.” 

And next on p. 18: “During peer-review, a third study involving a meta-analysis of the VR based on a cohort of 28 industry-sponsored trials of selective serotonin reuptake inhibitors (SSRIs) for depression was published online [52]. The study, which was not based on a pre-registered protocol, in accordance with our findings, found no difference in the variance of the HAMD-17 endpoint scores between patients receiving SSRIs and patients receiving placebo. Taken together, the similar findings in three other studies using overlapping but different study populations and various methodologies, may be taken as one indicator of the robustness of our findings.”

Response: We have checked that our manuscript meets PLOS ONE’s style requirements and made few minor edits in that regard. 

2. We noted in your submission details that a portion of your manuscript may have been presented or published elsewhere.

"An earlier version of the manuscript has been previously published as a preprint in Psyarxiv (doi: 10.31234/osf.io/m4aqc).

The present manuscript is not currently under for publication by any other journal and has not been published in any other form elsewhere."

Please clarify whether this publication was peer-reviewed and formally published. If this work was previously peer-reviewed and published, in the cover letter please provide the reason that this work does not constitute dual publication and should be included in the current manuscript.

Response: While our manuscript has been posted as a preprint in PsyArXiv (without peer-review), it has not previously been peer-reviewed and formally published.

Comments from the reviewers

Reviewer #1

This is a well written and timely study. Although the research question was previously addressed by two other groups (Volkmann et al; Plöderl and Hengartner) I think this study is important as, first, it adds an insightful simulation analysis, and second, consistently replicates the findings of the previous studies. Having that said, I want to declare that I am one of the authors of these studies (M. Hengartner).

I have a few suggestions:

The authors may want to add on page 5 that the search for biomarkers and clinical predictors of differential treatment response has consistently failed thus far, which is clearly at odds with the prevailing belief that patients respond differently to antidepressants. Given that equal variance ratios do not exclude the possibility that treatment effect heterogeneity exists, as the authors also rightly state in the discussion, the lack of robust treatment effect modifiers, despite decades of research and millions spend on the search for it, speaks volumes in my opinion. The persistent claim that differential treatment response exists should be supported by evidence. Given that there is no consistent evidence for this claim, it would be more sensible to assume that there is no treatment effect heterogeneity, unless proven otherwise. That so many people firmly stick to this view probably indicates that most researchers and physicians conflate observed treatment outcomes with treatment effects.

Response: Thank you for this very relevant suggestion. We have now added the following sentence on page 5 to inform of the lack of clinical characteristics and biomarkers to predict antidepressant response: “Despite decades of research, however, no clinical characteristic and no individual or aggregate biomarker [30] has translated into clinical practice for the guidance of treatment selection.”

The study by Maslej et al (ref 36) was retracted after we pointed out in a letter to the editor that their analysis was flawed. See the retraction note here (https://jamanetwork.com/journals/jamapsychiatry/fullarticle/2767242) and our letter to the editor here (https://www.researchgate.net/publication/342083346_Commentary_on_Maslej_et_al_No_evidence_of_individual_differences_in_response_to_antidepressants). This reference thus needs to be removed.

Response: The study by Maslej et al. was indeed retracted after submission of our manuscript and we have now removed references to the study throughout our manuscript. 

The authors claim that Volkmann et al as well as Plöderl and Hengartner did not address important methodological issues. I assume that they refer to the simulation study included in their manuscript. If so, they should specify. Otherwise, I don’t really see which methodological issues Munkholm et al addressed that were not addressed in the other two studies. In fact, in my view Volkmann et al. addressed various methodological issues that Munkholm et al. did not consider. For instance, Volkmann et al. empirically addressed the important question, why the VR is the more appropriate method than the CVR, which was erroneously applied in the now retracted paper by Maslej et al. Volkmann et al also empirically tested, how likely treatment-by-patient interaction would be if VR=1.

Response: Our statement in the introduction that previous studies did not address important methodological issues pertained to the consideration of assumptions and requirements regarding the analysis of the CVR. Thus, we address in detail in our discussion the nature of the scales used to assess depression symptom severity and the implications for analysis of the CVR, which was not considered in previous studies. We feel that a specification of this in the introduction, as suggested by the reviewer, would make the introduction section a bit too technical and have revised the sentence in the introduction section, p. 6, which now reads: “Two recent studies applied similar methods and found no evidence of heterogeneous treatment effects of antidepressants for depression [37,38]; one of those studies also provided a helpful discussion of the variability ratio as an indicator for treatment effect variability [38]”

We agree that the approach employed by Volkmann et al. when analyzing the CVR was more appropriate than that of Maslej et al., as their analysis considered the correlation between mean outcome values and the variance. However, given the issues related to the scales we discuss, we believe it is not relevant to our study.

Shortly after publication of our analysis (Plöderl and Hengartner), we were attacked by Fredrik Hieronymus on Twitter that our study, and by consequence also the present study by Munkholm et al., was severely flawed as it was based on trial-level data instead of individual-patient data. A few days ago, Hieronymus et al now published their own analysis based in IPD data (“Individual variability in treatment response to antidepressants in major depression: comparing trial-level and patient-level analyses”; Acta Psychiatrica Scandinavica, doi: 10.1111/ACPS.13205). To be honest, I don’t really know what to make of the Hieronymus study, as their methodological approach seems rather arbitrary and was not prespecified in a protocol. They corrected the placebo endpoint scores to match the mean scores in the active group and then compared the two distributions with a Kolmogorov-Smirnov test, which, as is well established, will almost always yield a statistically significant result when sample size is large. I also wonder how valid it is to artificially change the placebo endpoint scores, as there is no clear rational, what endpoint score an individual placebo recipient would have achieved had he/she received the antidepressant. The approach chosen by Hieronymus et al is just one possible option among many, and a different approach of course would have produced a different distribution. In any case, I suggest Munkholm et al. have a critical look at this paper and comment on it in the manuscript.

Response: Thank you for directing our attention to the paper by Hieronymus et al. We noted that the study by Hieronymus et al. was non-transparently reported, was not based on a pre-registered protocol and the authors did not share their data or analysis code. Based on analysis of their dataset of 28 industry-sponsored trials they did not find any difference in VR between patients receiving one of three SSRIs or placebo, in accordance with our findings and those of Volkmann et al. and Plöderl et al. The authors also compared the distributions of the endpoint depression severity scores between the two groups and concluded that their finding of a statistically significant difference suggest the presence of individual differences in response to antidepressants. However, as we detail in our manuscript, RCTs cannot demonstrate the presence of a treatment-by-patient interaction. As the focus of our study was the investigation of the variability ratio, we do not find their latter finding to be relevant to our study, but now mention their finding regarding the variability ratio in the discussion, p. 18: “During peer-review, a third study involving a meta-analysis of the VR based on a cohort of 28 industry-sponsored trials of selective serotonin reuptake inhibitors (SSRIs) for depression was published online [52]. The study, which was not based on a pre-registered protocol, in accordance with our findings, found no difference in the variance of the HAMD-17 endpoint scores between patients receiving SSRIs and patients receiving placebo. Taken together, the similar findings in three other studies using overlapping but different study populations and various methodologies, may be taken as one indicator of the robustness of our findings.” .

On page 17 the authors discuss in detail the CVR. They make legitimate points, but in my view the main issue why the CVR is inappropriate is the assumption of linear association between the natural logarithm of the mean and the natural logarithm of the standard deviation with a slope coefficient of 1. As comprehensively shown by Volkmann et al., this assumption is severely violated, since the slope coefficient is only about 0.1. The VR assumes a slope coefficient of 0, thus has a much better fit to the data than the CVR. This should be added to the text.

Response: Thank you for this comment. While we think that the assumptions regarding the nature of the scales analyzed is the primary issue related to the dataset used in our study, we agree that the failure to address the correlation between the mean and the standard deviation if analyzing the CVR in cases where these assumptions are met, is important. We have added this to the discussion, p. 17: “Another issue related to the CVR is the necessity of addressing the relationship between the mean and the standard deviation in the analysis, as spurious findings may otherwise be observed [38].”.

Reviewer #2

In the manuscript “Individual response to antidepressants for depression in adults - a meta-analysis and simulation study”, Munkholm et al. conducted a simulation study and inverse variance random-effects meta-analysis in order to compare the outcome variance in patients receiving antidepressants with the outcome variance in patients receiving placebo in randomized controlled trials of adults with major depressive disorder, as an indicator of individual differences in response to antidepressants. The authors found that there were no differences in variability ratios between across antidepressant vs. placebo comparisons, a result that indicates that the null hypothesis of equal variances cannot be rejected. In the opinion of this reviewer, the authors’ description of methods and results are comprehensive and the paper as a whole a strong contribution to the literature. The research has many strong points, such as: a) a pre-registered protocol, b) the use of a comprehensive dataset from which 222 studies comparing antidepressants vs placebo were extracted, c) all data and code are openly shared on the Open Science Framework, d) the use of relative variability as an indicator for treatment-by-patient interaction, e) conducting a simulation study to investigate possible components of variation in randomized controlled trials and, many more. The results of the simulation study are especially informative.

On a final note, although the database that was used both for informing the simulation study and to conduct the meta-analysis is comprehensive and encompasses many trials, the search for RCTs ended in 2016. As a personal curiosity, although it is clearly stated in the protocol that no supplementary data would be used, it would be very interesting how the results would change if the authors actualized the search for the period 2016-2020 and integrated those results also.

Response: We thank the reviewer for the very positive comments regarding our study. We agree that it could be of interest to include recent studies through an updated search but, as noted by the reviewer, this was beyond the scope of our study. We have added this as a limitation to the discussion section, p. 18-19: “Fifth, the search for studies included in the dataset was conducted in 2016. However, given our findings and the large dataset we used, it could be considered unlikely that adding expectedly relatively few studies through an updated search would change our results.”.

---

## [Decision Letter · Decision Letter 1]

6 Aug 2020

Individual response to antidepressants for depression in adults - a meta-analysis and simulation study

PONE-D-20-17299R1

Dear Dr. Munkholm,

We’re pleased to inform you that your manuscript has been judged scientifically suitable for publication and will be formally accepted for publication once it meets all outstanding technical requirements.

Kind regards,

Florian Naudet, M.D., M.P.H., Ph.D.

Academic Editor

PLOS ONE

Additional Editor Comments (optional):

Reviewers' comments:

Reviewer's Responses to Questions

**Comments to the Author**

1. If the authors have adequately addressed your comments raised in a previous round of review and you feel that this manuscript is now acceptable for publication, you may indicate that here to bypass the “Comments to the Author” section, enter your conflict of interest statement in the “Confidential to Editor” section, and submit your "Accept" recommendation.

Reviewer #1: All comments have been addressed

2. Is the manuscript technically sound, and do the data support the conclusions?

Reviewer #1: Yes

3. Has the statistical analysis been performed appropriately and rigorously? 

Reviewer #1: Yes

4. Have the authors made all data underlying the findings in their manuscript fully available?

Reviewer #1: Yes

5. Is the manuscript presented in an intelligible fashion and written in standard English?

Reviewer #1: Yes

6. Review Comments to the Author

Reviewer #1: (No Response)

7. PLOS authors have the option to publish the peer review history of their article (what does this mean?). If published, this will include your full peer review and any attached files.

Reviewer #1: **Yes: **Michael P. Hengartner

---

## [Editor Report · Acceptance letter]

11 Aug 2020

PONE-D-20-17299R1 

Individual response to antidepressants for depression in adults - a meta-analysis and simulation study 

Dear Dr. Munkholm:

I'm pleased to inform you that your manuscript has been deemed suitable for publication in PLOS ONE. Congratulations! Your manuscript is now with our production department. 

Kind regards, 

on behalf of

Pr. Florian Naudet 

Academic Editor

PLOS ONE